Low-frequency, low-magnitude vibrations (LFLM) enhances chondrogenic differentiation potential of human adipose derived mesenchymal stromal stem cells (hASCs)

Marycz Krzysztof 1 2 krzysztofmarycz@interia.pl
Lewandowski Daniel 3
Tomaszewski Krzysztof A. 4 5
Henry Brandon M. 4
Golec Edward B. 5 6
Marędziak Monika 7
1 Faculty of Biology, University of Environmental and Life Sciences , Wroclaw , Poland
2 Wroclaw Research Centre EIT + , Wroclaw , Poland
3 Department of Mechanics, Materials Science and Engineering, Wrocław University of Technology , Wrocław , Poland
4 Department of Anatomy, Jagiellonian University Medical College , Krakow , Poland
5 Department of Orthopaedics and Trauma Surgery, 5th Military Clinical Hospital and Polyclinic , Krakow , Poland
6 Faculty of Motor Rehabilitation, Bronislaw Czech University School of Physical Education , Krakow , Poland
7 Faculty of Veterinary Medicine, Department of Animal Physiology and Biostructure, University of Environmental and Life Sciences , Wroclaw , Poland
Rehman Jalees
Electronic publication date: 2016 Feb 25
Publication date: 2016
Volume: 4
Electronic Location ID: e1637
Received 2015 Nov 17; Accepted 2016 Jan 7
Copyright: ©2016 Marycz et al.
Copyright year: 2016
Copyright holder: Marycz et al.
License: This is an open access article distributed under the terms of the Creative Commons Attribution License, which permits unrestricted use, distribution, reproduction and adaptation in any medium and for any purpose provided that it is properly attributed. For attribution, the original author(s), title, publication source (PeerJ) and either DOI or URL of the article must be cited.
License URL: https://creativecommons.org/licenses/by/4.0/

Keywords: Low-magnitude low-frequency vibration stimulation, Chondrogenesis, Adipose-derived mesenchymal stem cells, Adipogenesis

Funding: Wroclaw Research Centre EIT European Regional Development Fund POIG.01.01.02-02-003/08 Wrocław Centre of Biotechnology Foundation for Polish Science (FNP) The research was supported by Wroclaw Research Centre EIT + under the project ‘Biotechnologies and advanced medical technologies’—BioMed (POIG.01.01.02-02-003/08) financed from the European Regional Development Fund (Operational Programmed Innovative Economy, 1.1.2.). This publication was supported by Wrocław Centre of Biotechnology, the Leading National Research Centre (KNOW) program between 2014 and 2018. Krzysztof A. Tomaszewski was supported by the Foundation for Polish Science (FNP). The funders had no role in study design, data collection and analysis, decision to publish, or preparation of the manuscript.

==============================
The aim of this study was to evaluate if low-frequency, low-magnitude vibrations (LFLM) could enhance chondrogenic differentiation potential of human adipose derived mesenchymal stem cells (hASCs) with simultaneous inhibition of their adipogenic properties for biomedical purposes. We developed a prototype device that induces low-magnitude (0.3 g) low-frequency vibrations with the following frequencies: 25, 35 and 45 Hz. Afterwards, we used human adipose derived mesenchymal stem cell (hASCS), to investigate their cellular response to the mechanical signals. We have also evaluated hASCs morphological and proliferative activity changes in response to each frequency. Induction of chondrogenesis in hASCs, under the influence of a 35 Hz signal leads to most effective and stable cartilaginous tissue formation through highest secretion of Bone Morphogenetic Protein 2 (BMP-2), and Collagen type II, with low concentration of Collagen type I. These results correlated well with appropriate gene expression level. Simultaneously, we observed significant up-regulation of α3, α4, β1 and β3 integrins in chondroblast progenitor cells treated with 35 Hz vibrations, as well as Sox-9. Interestingly, we noticed that application of 35 Hz frequencies significantly inhibited adipogenesis of hASCs. The obtained results suggest that application of LFLM vibrations together with stem cell therapy might be a promising tool in cartilage regeneration.

Introduction

Articular cartilage injuries are a growing problem in both human and veterinary medicine. Injury to cartilage manifests through the typical signs of inflammation, and can be caused by either trauma or diseases such as osteonecrosis, cartilage necrosis, or arthritis. Because cartilage is an avascular tissue with chondrocytes that are characterized by a low mitotic potential, the regenerative potential cartilage is substantially limited (Chung & Burdick, 2008). As such, the spontaneous regeneration of injured cartilage is extremely difficult. Until recently, the vast majority of the available treatment methods have focused on eliminating symptoms and improving patient quality-of-life through the use steroidal or non-steroidal anti-inflammatory drug treatment (NSAIDs) (Lin et al., 2004). However, when used long-term, these medications may lead to chondronecrosis (Brandt, 1987).

A potential solution to this problem emerges in the form of cell based therapies. Adult mesenchymal stem cells (MSCs) may be a possible source of cells for this type of therapy due to their immunomodulatory action, ability to self-renew, and ability to differentiate into several cell lineages, i.e., chondrocytes, osteoblasts or adipocytes (Iyer & Rojas, 2008; Zuk et al., 2001). Currently, bone marrow (BMMSCs) and adipose derived mesenchymal stem cells (ASCs) are the cells most frequently applied in cell-based therapies at the preclinical stage. Of the two cell types mentioned above, ASCs seem a better alternative to BMMSCs, due to their easy accessibility, and thus lower donor-related risks (Baer & Geiger, 2012). Moreover, activated ASCs secrete from their surface small, spherical membrane fragments called microvesicles (MVs) (Marędziak et al., 2015). These MVs contain important regenerative molecules, that improve the function of damaged tissues—eg., growth factors, bioactive lipids, proteins. Microvesicles secreted by MSCs, stimulated to differentiate into osteocytes, release into the culture medium compounds rich in Collagen type I and II or Bone Morphogenetic Protein 2 (Collino et al., 2010; Tetta et al., 2012). Several studies have confirmed the beneficial clinical effect of ASCs in the treatment of musculoskeletal disorders, particularly in the field of veterinary orthopedics (Marycz et al., 2012; Marycz et al., 2012; Brittberg et al., 1994). In our previous study, we demonstrated the positive effects of ASCs application in equine and canine osteoarthritis treatment (Nicpoń et al., 2014).

However, one major limitation to the clinical application of ASCs is the age-related decrease in the proliferative and chondrogenic differentiation potential of ASCs obtained from older populations (Choudhery et al., 2014). As degenerative joint diseases increase in prevalence with age, it is important to develop methods to overcome this limitation. In order to improve both the proliferative and the differentiation potential of ASCs, various physical stimuli such as static magnetic field (Marędziak et al., 2014), electric signals (Hammerick et al., 2009) and cyclic strain (Simmons et al., 2003) have been applied. However, this field still remains relatively unexplored in terms of the biological effects of low-magnitude low-frequency vibration (LMLF)—a non-invasive biophysical intervention that leads to cyclic loading of the targeted tissue. LMLF vibrations include values of magnitude under 1.0 g, where g = 9, 81m∕s2, and a frequency between 20–100 Hz (Lau et al., 2010). Several studies have investigated the role of various magnitudes and frequencies of vibrations, such as high-magnitude low-frequency (HMLF) vibrations (Nikander et al., 2009), high-magnitude high frequency vibrations (HMHF) (Tirkkonen et al., 2011) and low-magnitude high-frequency (LMHF) vibrations (Luu et al., 2009), in the context of their influence on cellular response (Edwards & Reilly, 2015; Uzer et al., 2015; Sen et al., 2011; Prè et al., 2013; Uzer et al., 2013). Moreover, it has been reported that LMHF enhance the osteogenic differentiation potential of MSCs (Tirkkonen et al., 2011). Enhancement of osteogenic and/or chondrogenic differentiation potential of MSCs may strongly depend on up-regulation of particular integrins, that are activated by various biomechanical signals (Popov et al., 2015). Integrins are heterodimeric glycoproteins that are composed of an α- and a β-subunit, each of which has an extracellular and a cytoplasmic domain (Goessler et al., 2009). Several studies have provided evidence that chondrocytes express integrins (Hering, 1999; Hynes, 1992; Giancotti & Ruoslahti, 1999; Albelda & Buck, 1990; Salter et al., 1992; Lee, Qi & Scully, 2002). In particular, the α1β1 and α5β1 integrins have been shown to be the most prominent in adult chondrocytes isolated from normal articular cartilage. However, the other integrins are still poorly investigated, especially in the context of their expression in differentiated precursor cells additionally stimulated by various types of external mechanical or others signals.

In an animal model, LMHF signals had a positive influence on both bone formation and density, enhancing bone strength and recovery after bone fracture (Xie, Rubin & Judex, 2008; Wehrle et al., 2015; Rubin, Judex & Qin, 2006). Moreover, preliminary studies in children with disabling conditions and post-menopausal women indicate that such signals can be efficacious in reversing and/or preventing bone loss (Rubin, Judex & Qin, 2006). However, to the best knowledge of the authors, the current literature lacks data concerning the effects of LMLF vibrations on the chondrogenic differentiation potential of human ASCs.

The aim of this study was to investigate how harmonic vibration, sinusoidal with constant low-magnitude (0.3 g, where g = 9, 81m∕s2) and low-frequency (25, 35, 45 Hz) mechanical signals, generated by an actuating device, effects ASC morphology, growth, and adipogenic and chondrogenic differentiation potential.

Materials and Methods

Description of the cell vibration generator prototype

The process of inducing vibrations was applied using custom-made vibration platforms, specially constructed device that allowed to induce mechanical motion of a 24-well culture plate. Movement of the plate was characterized by the harmonic sine of a given amplitude and frequency. The direction of plate translations was perpendicular to the main surface on which the cells were cultured.

A scheme of the stand is shown in Fig. 1C and photographs in Figs. 1A and 1B. The vibration generator, an electromagnetic actuator, is positioned on a stationary base. The principle of its operation is based on coil movement, which is generated by an alternating current flow. The design of the actuator is similar to that of a typical loudspeaker, with the main difference being that the moving parts are not a flexible membrane but a stiff plate.

Figure 1 Vibration generation prototype.

Actuator with cell culture plate connected with PC software, put inside a CO2 incubator (A). Cell culture plate attatched to the spacer under electro-magnetic actuator (B). Connection diagram and flow of signals during vibration stimulation. The electromagnetic actuator was supplied directly by the amplifier. The displacement signal waveform was generated in the computer software and sent through a digital-to-analog converter to the amplifier. 1, incubator; 2, 24-well culture plate; 3, spacer; 4, stiff movable plate; 5, electro-magnetic actuator (EM-ACTUATOR) with coil inside; 6, signal amplifier; 7, measuring card with A/C and C/A converters; 8, PC and software; 9, laser displacement sensor head; 10, laser beam. x shows the movement direction of the culture plate (C).

The stiff plate of the actuator moves in a linear manner in relation to the stationary part. Fig. 1C depicts the displacement value as ‘x’. Between the actuator and the culture plate, a spacer has been mounted. The spacer is a rigid element made of polyethylene, placed to create distance from the cell culture actuator. This was done to eliminate the possible influence of the alternating magnetic field generated by the EM-actuator on the cell culture. The height of the spacer was about 10 cm. The strength of the magnetic field at this distance does not differ from the background.

The culture plate was attached to the top surface of the spacer in such a way to allow quick mounting. Such a method was dictated by the fact that the vibration stimulation was scheduled only for short periods of time each day. Movement of the culture plate was defined as a course of the sine function with a given value of frequency and amplitude of acceleration. A laser displacement sensor (KEYENCE LK-G157) was used to measure the translation of the culture plate. The acceleration signal was calculated according to the following formula: x=Asinωt→x ¨=−Aω2 sinωt

where x—displacement, x ¨—acceleration, A—amplitude of displacement, ω—frequency of vibrations (ω = 2πf), f—frequency, t—time, Aω2—amplitude of acceleration.

Vibration loading protocol of hASCs culture

The hASCs were seeded at a concentration of 3 × 104 on 24-well plates and 5 × 104 to a 15 ml tube. Tubes with 3D model were vibrated on tube rack. For each vibration model (25, 35 and 45 Hz) separate dishes were used. Plates/racks were placed securely onto the vibration device and oscillated vertically at 25, 35 and 45 Hz. The stimulus was sinusoidal and delivered with a peak acceleration of 0,3 g for 15 min once a day, for 14 consecutive days. Cells in the non-vibration group were placed on the same but stationary plate. After 15 min of vibration, the hASCs (both vibrated and non-vibrated groups) received fresh culture medium.

Isolation of human adipose derived mesenchymal stem cells (hASCs)

This study was approved by the local bioethics committee of Wroclaw Medical University, Poland (number KB-177/2014). Written informed consent was obtained from each patient prior to tissue collection during total hip arthroplasty. This study adhered to the Helsinki Declaration (1964) and its later amendments.

Subcutaneous adipose tissue was collected from 4 patients. From each patient we obtained 4 donor samples, representing of a total n = 16. The average age of the patients was 69 ± 1 years. Briefly, after collection, the tissue samples were placed in sterile Hank’s Balanced Salt Solution (HBSS). The isolation procedure of adipose-derived mesenchymal stem cells was conducted under aseptic conditions and in accordance with previously described protocol (Grzesiak et al., 2011; Marycz et al., 2013). Samples were washed with HBSS supplemented with a 1% antibiotic-antimicotic solution (penicillin/streptomycin/amphotericin b at a concentration of 0.017 mol/l, 0.01 mol/l and 0.0002 mol/l respectively; Sigma Aldrich, cat no A5955) and then cut into small pieces using surgical scissors. Next, the samples were placed in a sterile centrifuge tube and digested with type I collagenase (1 mg/ml, Sigma Aldrich, cat no C5894). After 30 min incubation at 37°C, the tissue homogenate was centrifuged at 1,200 g for 10 min. The supernatant was removed and the cell pellet was resuspended in growth media. The cell suspension was then transferred to the cell culture flask.

Immunophenotyping, Fluorescence-activated cell sorting (FACs) analysis, and multipotency test

Cells were plated on 24-well culture plates suspended in 500 µl of standard medium at a concentration of 8 × 103 cells per well. The presence of specific antigens for ASCs, i.e., integrin beta-1 (CD29), HCAM (CD44), 5′-nucleotidase (CD73) and endoglin (CD105) and leukocyte common antigen (CD45) was examined after one week of culture by means of primary antibodies (all from Sigma Aldrich). Negative staining of CD45 was used to exclude hematopoietic origin. After fixation, cells were permeabilized with 0.2% Tween 20 for 15 min and washed three times with HBSS. The solution of primary antibody and 4% FBS in PBS was applied to every well and incubated overnight at 4°C. Next, the cells were washed three times and secondary goat anti-rabbit conjugated with Atto 488 antibody was added to appropriate wells at concentration 1:150. After incubation at room temperature for 1.5 h, the cells were washed again and photographed under a fluorescence microscope.

For the multipotency test, cells were cultured on chondrogenic and adipogenic media (STEMPRO® Chondrogenesis/Osteogenesis Differentiation Kit and STEMPRO® Adipogenesis Differentiation Kit, Life Technologies) for 14 days. The culture media was changed every second day.

After 3 passages, the ASCs were examined for surface protein molecule expression by flow cytometry. Cells were trypinized using a Trypsin-EDTA solution (TrypLE™, Life Technologies), centrifuged at 400 xg for 3 min, and then washed with PBS containing 2% FBS (fetal bovine serum) (Sigma Aldrich). A total of 5 × 105 cells were labeled for 20 min (on ice and dark) with antibodies pre-conjugated with allophycocyanin (APC), peridinin chlorphyllprotein (PerCP), fluorescein isothiocyanate (FITC) or phycoerythrin (PE). The following CD surface markers were tested: CD34, CD45, CD105, CD90, CD73, CD44, CD29 and IgG1 as an isotype control antibody (BD Pharmingen). The samples were analyzed by a Becton Dickinson FACSCalibur flow cytometer. At least ten thousand events were acquired for each CD surface marker. The data was then analyzed using FlowJo X software (Treestar).

Cell culture

Throughout the experiment, hASCs were cultured in aseptic and constant conditions in an incubator at 37°C, 5% CO2 and 95% humidity. The cell population was plated in T-75 culture flasks for primary culture and was maintained in Dulbecco’s Modified Eagle’s Medium (DMEM) with nutrient F-12 Ham (Sigma Aldrich) supplemented with 10% FBS and 1% of antibiotic/antimicotic solution at a concentration of 0,017 mol/l, 0,01 mol/l and 0,0002 mol/l respectively (Sigma Aldrich, cat no A5955). The culture medium was changed every second day. Human ASCs were passaged using Trypsin-EDTA solution (TrypLE™, Life Technologies) in accordance with manufacturer’s instruction after reaching about 80–90% confluence. Cells were passaged three times before use in experiments.

Isolated adipose-derived mesenchymal stem cells were divided into 2 groups. The first one was stimulated with chondrogenic medium and the second one with adipogenic medium. The differentiation processes of hASCs were performed using the STEMPRO® Chondrogenesis Differentiation Kit and STEMPRO® Adipogenesis Differentiation Kit (Life Technologies), respectively. Cells were seeded at 3 × 104 on 24-well plates and 5 × 104 to a 15 ml tube. The stimulation of cells was performed in accordance with the manufacturers’ instruction.

The chondrogenic culture was maintained in two systems: 2D in 24-well plates for fluorescent, histochemical stainnings, rtPCR analysis and SEM, and 3D for ELISA tests and Focused Ion Beam Scanning Electron Microscope (FIB-SEM, Auriga Compact Crossbeam, Zeiss, Germany). For the 3D system, 2.5 × 105 hASCs were seeded into 15 ml polypropylene tubes and pelleted. The hASCs were cultured for 14 days as 3D pellets in induction medium STEMPRO® Chondrogenesis Differentiation Kit (Life Technologies). The experiment was repeated three times. Both 2D and 3D cultures were incubated two days before starting vibration stimulation.

Cell proliferation assay

The cell proliferation factor (PF) was evaluated using the Alamar Blue test (TOX-8, Sigma Aldrich) according to the manufacturer’s instructions. The culture media was replaced with a medium containing 10% of resazurin-based dye and incubated for two hours. Afterwards, the supernatants were collected and subjected to absorbance measurement by means of spectrophotometer (SPECTRO StarNano, BMG Labtech) at 600 nm of wavelength, with a distraction of 690 nm of background absorbance. The procedure was performed during the differentiation period at days 2, 5, 10, and 14.

A standard curve obtained during the experiment, allowed to estimate the amount of cells. The population doubling time (PDT) was assessed using an online calculator (http://www.doubling-time.com/compute.php).

Examination of hASCs morphology

Cell morphology, cellular composition, and culture growth pattern were analyzed using an inverted, fluorescence microscope (AxioObserverA1, Zeiss) and a scanning electron microscope (SEM; EVO LS15, Zeiss).

In order to begin observations, after the culture period, cells were fixed with 4% paraformaldehyde. After 15-min permeabilization with 0.2% Tween, cells were stained using atto-565-labeled phalloidin for 40 min to visualize actin filaments. After triple washing, diamidino-2-phenylindole (DAPI) staining was applied for 5 min to analyze the distribution of cell nuclei. All fluorescence staining was performed at room temperature in the dark. Additionally, Oil Red O staining for adipogenic cultures and Safranin-O staining for chondrogenic cultures were performed to observe fat droplets and chondrocytes, respectively. After fixation, adipogenic samples were treated with 0.1% Oil Red O solution for 5 min, Plates were rinsed 3×with water and images of cells on plate were taken. For quantification the percentage of Oil Red O absorption, the dye was extracted by isopropanol and absorbance was determined at 490 nm. Chondrogenic samples were treated with 1% acetic acid for 10 min and stained with Safranin-O for 5 min. Images were acquired using a Cannon PowerShot digital camera. In order to evaluate the chondrogenic differentiation efficiency, the concentration of proteoglycans was determined, basing on the binding Safranin O to glycosaminoglycans and spectrophotometric measurements at a 470 nm wavelength and assessment percentage of Safranin O absorption. As 100% control, we adopted reagent not added to culture. To analyze detailed morphological features of the cells, especially fat droplets, and chondrogenic nodules, SEM was performed. After fixation, the cells were washed in distilled water and dehydrated in ethanol (concentrations from 50 to 100%, every 5 min). Thoroughly dried cells were coated with gold (ScanCoat 6, Oxford), placed in a microscope chamber, and observed using the SE1 detector, at 10 kV of filament’s tension. To observe morphological features and measure diameters of nodules (n = 6) Ion Beam Scanning Electron Microscope (FIB-SEM, Auriga Compact Crossbeam, Zeiss, Germany) obsrvations were performed. Analysis was performed using a focus ion beam detector at magnification of 200X. Diameter of adipocytes (n = 6) was measured using Scanning Electron Microscope (SEM; EVO LS15, Zeiss).

Enzyme-linked immunosorbent assays

In order to evaluate the chondrogenic differentiation efficiency on the protein level, the concentration of chondrogenesis-specific markers was investigated. The total concentration of proteins from pellet cultures was determined with enzyme-linked immunosorbent assay (ELISA). For the analysis, cells homogenate were collected on the last day of the experiment. Chondrogenic media was subjected to a BMP-2 ELISA assay (Bone Morphogenic Protein 2 Quantikine ELISA Kit, R & D Systems), and a Col-1 and a Col-2 ELISA assay (Human Collagen alpha-1(I) and (II) chain ELISA Kit, EIAab). All steps of each ELISA tests were performed in accordance with the manufacturer’s protocol. Each sample was prepared in duplicate. Spectrofotometric determination was performed using a microplate reader (Spectrostar Nano, BMG Labtech) at a wavelength equal to 450 nm and with the correction wavelength of 540 nm. The concentration of proteins was presented as a ratio of protein weight and supernatant volume (w/v).

Quantitative real-time reverse transcription polymerase chain reaction (qRT-PCR)

In order to analyze gene expression, cells after stimulation were rinsed with HBSS and were homogenized with 0.5 ml of TRI Reagent (Sigma Aldrich) directly in the culture well. The total RNA was isolated using a phenol-chloroform method as previously described (Chomczynski & Sacchi, 1987). After isolation, total RNA was diluted in DEPC-treated water. The concentration and purity of RNA preparations was determined by absorbance measured at 260 nm with a nanospectrophotometer (VPA biowave II). Preparation of DNA-free RNA was performed using DNase I RNase-free kit (Thermo Scientific). For each reaction, 100 ng of total RNA was used. Transcription of gDNA-free total RNA to a complementary DNA (cDNA) was reverse transcribed using Moloney Murine Leukemia Virus Reverse Transcriptase (M-MLV RT) and oligo(dT)15 primers (Novazym). RNA purification and cDNA transcription was performed according to manufacturers’ instructions. Quantitative Real-Time polymerase chain reaction (qRT-PCR) was performed using 5 µl of cDNA in total volume of 20 µl by means of SensiFast SYBR & Fluorescein Kit (Bioline). The reaction was performed at a 500 nM final concentration of primers. The primer sequences used are presented in Table 1. qRT-PCR was performed as described previously (Kim & Im, 2010). Expression levels of all analyzed genes were normalized for the expression level of glyceraldehyde-3-phosphate dehydrogenase (GAPDH), a housekeeping gene.

Table 1 Sequences of qPCR primers.

Sequences of qPCR primers used for the amplification of human mRNA to chondrogenic genes.

Gene name	Primer sequentions	Ann. T, °C	Accession number	
GAPDH	Forward 5′-GTCAGTGGTGGACCTGACCT-3′	60	NM_002046	
Reverse 5′-CACCACCCTGTTGCTGTAGC-3′	
Collagen type I (COL1A1)	Forward 5′-GTGATGCTGGTCCTGTTGGT-3′	60	NM_000088.3	
Reverse 5′-CACCATCGTGAGCCTTCTCT-3′	
Collagen type II (COL2A1)	Forward 5′-GACAATCTGGCTCCCAAC-3′	60	NM_001844.4	
Reverse 5′-ACAGTCTTGCCCCACTTAC-3′	
Aggrecan (ACAN)	Forward 5′-GCCTACGAAGCAGGCTATGA-3′	60	NM_13227.3	
Reverse 5′-GCACGCCATAGGTCCTGA -3′	
SOX-9	Forward 5′-AGCGAACGCACATCAAGAC-3′	65	NM_000346	
Reverse 5′-GCTGTAGTGTGGGAGGTTGAA-3′	
RUNX-2	Forward 5′-GTGATAAATTCAGAAGGGAGG-3′	65	NM_001024630	
Reverse 5′-CTTTTGCTAATGCTTCGTGT-3′	
Collagen type X (Col-X)	Forward 5′-CAGTCATGCCTGAGGGTTTT-3′	65	NM_000493	
Reverse 5′-GGGTCATAATGCTGTTGCCT-3′	
Adiponectin (ADIQ)	Forward 5′-AGGGTGAGAAAGGAGATCC-3′	60	XM_011513324.1	
Reverse 5′-GGCATGTTGGGGATAGTAA-3′	
Leptin (LEP)	Forward 5′-ATGACACCAAAACCCTCATCAA-3′	60	XM_005250340.3	
Reverse 5′-GAAGTCCAAACCGGTGACTTT-3′	
PPAR-gamma	Forward 5′-ATGACACCAAAACCCTCATCAA-3′	60	AB565476.1	
Reverse 5′-GAGCGGGTGAAGACTCATGTCTGTC-3′	
Integrin α3	Forward 5′-ATCTTGAGAGCCACAGTCA-3′	52	( NM_002204)	
Reverse 5′-cTGGGTCCTTCTTTCTAGTTC-3′	
Integrin α4	Forward 5′-AATGGATGAGACTTCAGCACT-3′	58	( NM_000885)	
Reverse 5′-CTCTTCTGTTTTCTTCTTGTAGG-3′	
Integrin α5	Forward 5′-ACTAGGAAATCCATTCACAGTTC-3′	52	( NM_002205)	
Reverse 5′-GCATAGTTAGTGTTCTTTGTTGG-3′	
Integrin αv	Forward 5′-GGAGCACATTTAGTTGAGGTAT-3′	56	( NM_002210)	
Reverse 5′-ACTGTTGCTAGGTGGTAAAACT-3′	
Integrin β3	Forward 5′-CTGCTGTAGACATTTGCTATGA-3′	52	( NM_000212)	
Reverse 5′-GCCAAGAGGTAGAAGGTAAATA-3′	
Integrin β5	Forward 5′-CTGTGGACTGATGTTTCCTT-3′	54	( NM_002213)	
Reverse 5′-GTATGCTGGTTTTACAGACTCC-3′	
Integrin β5	Forward 5′-GAAGGGTTGCCCTCCAGA-3′	60	NM_002211.3	
Reverse 5′-GCTTGAGCTTCTCTGCTGTT-3′	

Statistical analysis

All experiments were performed with at least 3 (n = 3) independent experiments (biological replicates, n ≥ 4) measured as quadriplicate or more (technical replicates, n ≥ 4).

Statistical analysis was performed using GraphPad Prism 5 software. The statistical significance of results was calculated using the one-way analysis of variance (ANOVA) with post-hoc Dunnett’s test by means. A P-value of less than 0.05 was considered statistically significant.

Results

hASCs—FACs analysis, immunophenotyping and multipotency test

Flow cytometry analysis revealed that hASCs showed positive labeling for CD29, CD44, CD73, CD105, and CD90 (Fig. 2B). The investigated cells were negatively labeled for two hematopoietic markers: CD34 and CD45 (Fig. 2B). Additionally, immunohistochemical staining confirmed the presence of mesenchymal markers (CD29, CD44, CD73, CD105) and excluded hematopoietic origin (CD45) (Fig. 2A).

Figure 2 Phenotyping and multipotency test.

The expression of specific cell markers CD29, CD44, CD73 and CD105 and the lack of hematopoetic cell marker CD45 (A). Characterization of hASCs FACS analysis. FACS histograms of passage 3 ASC simultaneously stained for CD45, CD34, CD105, CD90, CD73, CD44, CD29, and IgG1 as negative control. Histograms are representative of 3 independent flow cytometry analyses. Red histograms: IgG1 negative control; blue histograms: antibody specific staining (B). Multipotency assay- standard culture and differentiated cultures after Alizarin Red staining for osteogenic stimulation (mag. 50×, scale bar = 200 µm) and Oil Red O staining for adipogenic stimulation (mag. 100×, scale bar = 400 µm) (C).

Moreover, the multipotent character of hASC’s was confirmed by abundant osteogenic and adipogenic differentiation. In contrast to cells cultured under control conditions, the presence of osteo nodules, was observed in hASCs cultivated under osteogenic conditions (Fig. 2C). Moreover, mineral calcium deposits visualized by Alizarin Red staining were clearly detected after 3 weeks of osteogenic differentiation. After 2 weeks of culture with an adipogenic inducing media, hASCs developed Oil Red O positive lipid droplets (Fig. 2C), whereas control cultures grown in standard media failed to produce similar results (Fig. 2C).

Proliferation rate (PF) and population doubling time (PDT) of 3D chondroblasts originated from chondro induced hASCs

The proliferative activity, as well as the PDT, was analyzed during the 14 days of hASCs culturing in chondrogenic induction medium exposed to vibration frequencies of 25, 35 and 45 Hz and in non-vibration control conditions. The obtained data showed that all investigated vibration frequencies influenced the proliferative potential, as well as the PDT, of chondroblasts originated from hASCs (Figs. 3A and 3B). The percentage of Alamar Blue reduction decreased proportionally with cell count and activity.

Figure 3 Chondrogenesis proliferation factor and population doubling time.

Proliferation factor (A) and population doubling time (PDT) (B) of hASCs treated with 0, 25, 35 and 45 Hz vibration frequencies during chondrogenic stimulation. ∗p-value < 0.05.

Cells cultured under 25 Hz frequency reached the highest PF after 2 days of incubation and then declined, but on the 10th day of culture reached higher PF and declined again till the 14th day. Moreover, we noticed the longest PDT (277 ± 25 h) when cells were stimulated with a 25 Hz frequency. The cells exposed to 35 Hz vibrations had a higher PF than control cultures at all investigated time points (Fig. 3A). During the analysis, hASCs after chondrogenic differentiation using 25 Hz vibrations resulted in the lowest proliferation potential, as well as longest PDT, when compared to the other investigated groups.

The morphology of chondroblasts originated from hASCs

After 14 days, formation of chondroblast-specific nodules could be clearly observed (Figs. 4G–4V and 5A–5G) and showed a strong orange signal after Safranin O staining (Figs. 4O–4R). Characteristic chondrocyte-like cells were observed in all of the investigated cultures, however, the cells cultured under the influence of 35 Hz vibrations more efficiently induced chondrodifferentiation (Fig. 4T). In these samples, nodules had the largest size (Fig. 6A) as well as exhibited the highest absorption of Safranin O (Fig. 6B). Cells cultured under 25 Hz vibrations absorbed less Safranin O dye and formed nodules with a significantly smaller diameter than samples cultured with 35 Hz vibrations (Figs. 6A and 6B). Chondroinduction of cells treated with 45 Hz were comparable to the control group. Although chondro-nodules had similar diameters, the absorption of Safranin O by cells cultured with 45 Hz frequencies was significantly higher (Figs. 6A and 6B).

Figure 4 qPCR and morphology chondroblasts.

(A) RT-qPCR for chondrogenesis genes: SOX9, COL-X, COL-2, RUNX2, COL-1 and ACAN from hASCs that underwent chondrogenic induction on culture plates with treatment with 0, 25, 35, 45 Hz vibrations. ∗p-value < 0.05 (B) Cell morphology of chondroblasts originated from hASCs cultured on plates visualized by fluorescence stainings (DAPI A-D and Phalloidin E-H), Safranin staining (I-L). Scale bars A-H 100 µm and I-L 200 µm and scanning electron microscope photographs (Mag. 2000×).

Figure 5 ELISA and chondro nodules.

(A) Comparison of Col-1 , Col-2 and BMP-2 levels by ELISA hASCs pellets after 14 days of chondro-induction. (B) Morphological characterization and comparison of chondro-nodules from cells cultured at pellet.

Figure 6 Safranin absorbtion and diametres of chondronodules.

Percentage of Safranin staining absorption (A) and the average diameter of chondrogenic nodules (B) from chondroblasts that originated from hASCs. ∗p-value < 0.05.

Quantitative Collagen 1 and 2 (Col-1, Col-2) and Bone Morphogenetic Protein 2 (BMP-2) assay and chondrogenic gene expression analysis (SOX-9, Col-X, Col-II, Runx, Col-I, ACAN)

The performed analysis showed an increase in collagen type 2 concentration in comparison to the amount of collagen type I in all groups where vibrations were applied (Figs. 5A and 5B). These results were additionally confirmed by positive chondrogenic differentiation of hASCs.

The highest difference on mRNA level between collagen type II and type I was observed in the cultures stimulated with 25 and 35 Hz vibrations, and in the control culture. The lowest concentration of collagen type II with respect to collagen type I was observed when 45 Hz frequency vibrations were applied (Figs. 5A and 5B). Cells treated with 35 Hz frequency vibrations tended to secrete significantly higher amounts of BMP-2 in comparison to the other groups (Fig. 5C). Exposure to the 25 Hz stimulation model resulted in secretion of lower a concentration of BMP-2 in comparison to the 35 Hz, however significantly higher when compared to the 45 Hz frequencies, as well as to the control culture.

The quantitative evaluation of the concentration of collagen type I and type II was additionally confirmed by gene expression analysis (Figs. 4C and 4E). The highest activity of collagen type II and its predominance over expression of collagen type I was observed in cells treated with 35 Hz vibrations. Similarly to the quantitative evaluation, exposure to 25 Hz vibrations resulted in lower expression of Collagen type II when compared to the 35 Hz vibrations model. The gene expression of SOX-9 and Col-X, the master transcription factors of chondrogenesis, gradually increased in 35 Hz treated cells compared to control group (Figs. 4A and 4B). Aggrecan (ACAN) and RUNX2, another chondrogenic markers, significantly increased after 25 Hz treatment (Figs. 4D and 4F).

Analysis of integrin expression in response to vibration stimulation

In order to sense and translate the applied external mechanical signals, cells express mechanoreceptors on their surface, such as integrins. In our study, we analyzed the expression changes of four alpha (α3, α4, α5 and αV) and three beta (β1, β3 and β5) integrin subunits (Fig. 7). qPCR analysis demonstrated a slight increase of integrin α3, α4, β1 and β3 subunit expression after 25 Hz stimulation in comparison to control (0 Hz). We also found that when cells were stimulated with 35 Hz vibrations, hASCs significantly upregulated integrin α3, α4, β1 and β3 subunit. Interestingly, after 35 Hz stimulation, the highest increase in expression of the β3 intergrin was observed. With respect to integrin subunits α5, αV, and β5, expression levels were similar between to stimulated groups, however down-regulated as compared to control.

Proliferation factor and population doubling time of human adipocytes originated from adipo induced hASCs

The proliferation factor and PDT were determined in the various groups after 14 days of adipogenic induction. The stimulated cultures were characterized by an irregular proliferation rate (Fig. 8A). Stimulation with a 25 Hz frequency resulted in an increase of PF when compared with the other groups. The highest PF was on day 5, and from that point on a decreasing trend was observed (Fig. 8A). The 25 Hz frequency group also has the shortest PDT in comparison to other experimental groups (Fig. 8B).

Figure 7 Integrin expression (qPCR).

Integrin expression changes after mechanical stimulation of ASCs. Quantitative PCR analysis for integrin alpha 3, 4, 5, V and beta 1, 3, 5 subunits. ∗p < 0.05.

Figure 8 Adipogenesis proliferation factor and population doubling time.

Proliferation factor (A) and population doubling time (PDT) (B) of hASCs treated with 0, 25, 35 and 45 Hz vibration frequencies during adipogenic stimulation. ∗p-value < 0.05.

In the 35 Hz and 45 Hz treatment groups, proliferation remained decreased at days 2, 5 and 10 of culture in comparison to the control and 25 Hz group, although this difference was not as pronounced as that on day 14. These results were also reflected in the PDT calculations—the 35 Hz and 45 Hz cultures had longer time to achieving PDT (Fig. 8B).

Figure 9 qPCR and morphology of adipocytes.

(A) RT-qPCR for adipogenesis genes: PPAR-gamma, ADIQ, LEP from hASCs that underwent adipogenic induction on with treatment with 0, 25, 35, 45 Hz vibrations. ∗p-value < 0.05 (B) Cell morphology of adipocytes originated from hASCs visualized using fluorescence stainings (Phalloidin, DAPI; Mag. 50×, scale bar 200 µm), Oil Red O Staining (100×, scale bar = 400 µm) and scanning electron microscope photographs (1,000×).

Morphology of ASCs after adipogenic differentiation

Microscopic analysis of adipocytes that originated from hASCs revealed that the 25 Hz vibrations mostly enhanced the adipogenic differentiation in comparison to the other groups (Fig. 9L). The findings from gene expression analysis showed noticeable increase of PPAR-γ and adiponectin (ADIQ) (Figs. 9A and 9B) in 25 and 35 Hz stimulated groups compared to control. However, qRT-PCR findings also showed similar gene expression of leptin (LEP) (Fig. 9C). The adipocytes derived from 25 Hz stimulated group were highly abundant with large lipid vacuoles, positively stained by Oil Red O staining, and absorbed the highest percentage of dye (Fig. 10A). The smallest adipocyte diameter was observed in the control group (ranges between 60 and 92 um), while adipocytes treated with 35 Hz were characterized by the highest average size (ranges between 90 and 119 um) (Fig. 10B).

Figure 10 Oil Red O absorption and adipocytes diameters.

Percentage of Oil Red O staining (A) absorption and the average diameter (B) of adipocytes. ∗p-value < 0.05.

Discussion

Cartilage defects, especially in cases of osteoarthritis, have a serious impact on the patient’s quality-of-life and functionality. The increasing prevalence of degenerative joint diseases is explained by the increasing life expectancy of the general population (Raeissadat et al., 2013). Tissue-engineering approaches including the application of externally applied signals such as stimulation with electric currents (Ciombor & Aaron, 2005; Foley et al., 2008), laser (Miloro, Miller & Stoner, 2007), or ultrasound vibration (El-Mowafi & Mohsen, 2005; Taylor et al., 2007), are promising tools for overcoming this problem. However, these methods also have several shortcomings, such as emission of high temperature or generation of rarefactional pressure, which may lead to mild heating, coagulative necrosis, tissue vaporization or inducing pulsation of pre-existing gas bodies (Miller et al., 2012).

Even more promising is the combination of MSCs therapy with innovative devices that are able to induce particular external signals, which enhance the regeneration of injured tissues. Since stem cells viability, proliferation status, and differentiation potential are widely connected with their regenerative potential (Krampera et al., 2006; Mishra et al., 2009), searching for external stimulating factors that may enhance the mentioned MSCs features, before clinical application seems to be a crucial factor, when MSCs for cartilage regeneration purposes are considered. Furthermore, effective external stimulation may help overcome the age-related decrease in chondrogenic differentiation potential of hASCs that currently poses a limitation to their clinical application in cell-based therapies (Choudhery et al., 2014). Therefore, in the current study, we hypothesized that LMLF may enhance the chondrogenic differentiation potential of hASCs and simultaneously alter the differentiation toward fat tissue.

We found that when the PF of hASCs cultured in chondrogenic conditioned medium is considered, both the 25 Hz and the 35 Hz vibration frequencies reduce the PF of hASCs. Additionally, the PDT reached the highest level in cells treated with 25 Hz, which strongly correlates with the obtained PF factor. As recently reported, the PDT of MSCs directly correlates with their replicative senescence, which is linked to the decrease in the size of cell aggregation (Yoon et al., 2011). In each experimental group, the PDT was higher in comparison with other studies (Hass et al., 2011). This may be due to the fact that cells were rapidly differentiated into osteoblasts or adipocytes, and that their proliferation activity was significantly reduced. Sepúlveda et al. (2014) reported that cell senescence abrogates the therapeutic potential of human mesenchymal stem cells in a lethal endotoxemia model. Therefore, searching for methods that might improve the PDT level seems to be crucial in the context of clinical application of MSCs. Interestingly, although hASCs treated with 35 Hz had low PF and long PDT, they also developed the largest chondro nodules. Furthermore, the highest concentration of absorbed Safranin O staining was observed in nodules originating from cells treated with 35 Hz vibration. This likely demonstrates that 35 Hz vibrations induce the highest synthesis of glycosaminoglycans (GAG), and thus indicates that the chondrogenic process is highly efficient.

The GAGs have an unquestionable influence on biomechanical properties of cartilage. They serve as an important component of extracellular matrix (ECM), which directly affects the integrity of cartilage. Our data demonstrates that the 35 Hz vibration model could be applied to stimulate chondrocytes, which originated from hASCs to produce ECM of high biomechanical properties manifested by rich in collagen type II and proteoglycans. The most extensive morphometrical properties, i.e., length and height of single nodules, were observed in 2D culture cells that were treated with the 35 Hz vibrations protocol. Slightly weaker chondro nodule development, as well as cytoskeleton formation, was observed in the 2D culture treated with 25 Hz vibrations in comparison to the others, as well as to the control group (Figs. 4G–4V).

In order to determine which of the investigated frequencies has a more significant impact on the process of chondrogenesis, we applied quantitative ELISA and qPCR in order to evaluate the concentration of BMP-2 and the relationship between collagen type I and II in the culture medium. We found in 3D culture treated with 35 Hz the highest concentrations of BMP-2 (Fig. 5), and at the same time -collagen type II, in the 2D culture treated with 35 Hz (Fig. 4C). However, 25 Hz vibrations caused an increase in synthesis, albeit slightly smaller amounts of investigated proteins. Our obtained results stand in good agreement with the findings of Cashion et al. (2014), who reported that low frequency vibrations can induce secretion of BMP-2 in human umbilical cord derived MSCs. It is worth noting that both the 25 Hz and 35 Hz vibration stimulated chondrocytes originating from hASCs had elevated synthesis of type II rather than type I collagen. It is well known that elasticity of articular cartilage is dependent on the proper relationship between synthesis and secretion of collagen types I and II. Simultaneously, we observed, that 35 Hz vibration stimuli in 2D culture caused up-regulation of sex-determining region Y protein (SRY)-box 9 (SOX9) that is the primary regulator of chondrogenic differentiation on an in vitro as well as an in vivo level. The observed up-regulation of Sox-9, strongly corresponded with elevated expression of collagen type II, which is to be expected as Sox-9 directly regulates expression of collagen gene (COLL II) in chondrogenic progenitor cells, as well as chondrocytes (Akiyama, 2008; Bell et al., 1997). Moreover, SOX9 determines functions of RUNX2 and exerts a dominant function over RUNX2 in mesenchymal precursors (Zhou et al., 2006). In this study, we confirmed this effect; gene expression of SOX9 and RUNX2 are overlapped and enhanced in cells treated with vibrations.

The positive effect of LFLM on the functional chondrogenic differentiation process might be explained by up-regulation of the integrin family. Here, we found that both frequencies i.e., 25 and 35 Hz, affect up-regulation of integrin α3, α4, β1 and β3 subunits on the mRNA level, although statistical significance was observed only in 35 Hz stimulated cells. Furthermore, β1 integrin has been shown to play a crucial role in attachment and survival of MSCs during the chondrogenesis process. Furthermore, we observed that LFLM, in general, significantly down regulates integrin α5β5 expression. Moreover, we observed down-regulation of α5 integrins of MSCs stimulated with all tested frequencies. Thus our results stand in good agreement with Martino and colleagues (2009), that showed involvement of α5 integrins more in osteogenic rather than chondrogenic differentiation process (Martino et al., 2009).

Finally, the effect of vibration loading with particular frequencies on the adipogenic differentiation potential of hASCs was investigated. We found that among all tested frequencies, 35 Hz significantly inhibited adipogenesis in hASCs. The obtained results stand in good agreement with the findings by Oh et al. (2011) who reported that 20 and 30 Hz subsonic vibrations inhibited the proliferation of 3T3-L1 preadipocytes. Additionally, we observed the smallest absorption of Oil red O staining when cells were treated with 35 Hz vibration. However we observed increased numbers of lipid droplets in the 25 Hz vibration group, but the observed changes were only on a morphological level. In adipogenic gene expression level, we did not observe significant changes between 25 Hz and that of other investigated groups.

Conclusions

In conclusion, the vibration-loading device designed for the purpose of this study successfully generated controlled vibrational forces to hASCs cultured on 24-well, as well as the 3D pelleted cell model. Our results indicate that LFLM vibrations act differently on both chondrogenic, as well as adipogenic potential of hASCs. The most important finding of this study suggests that 35 Hz frequency vibrations enhance chondrogenic potential of hASCs with simultaneous inhibition of hASCs differentiation toward adipocytes. Finally, we conclude that mechanical signals, especially 35 Hz frequency vibrations might be potentially used in construction of therapeutic devices which may prove useful in the field of articular degenerative diseases treatment.

Supplemental Information

Supplemental Information 1 SOX9 qPCR results

Click here for additional data file.

Supplemental Information 2 collagen X q pcr results

Click here for additional data file.

Supplemental Information 3 Col-2 qPCR results

Click here for additional data file.

Supplemental Information 4 RUNX qPCR results

Click here for additional data file.

Supplemental Information 5 Col-1 qPCR results

Click here for additional data file.

Supplemental Information 6 Aggrecan qPCR results

Click here for additional data file.

Supplemental Information 7 Col-1 ELISA results

Click here for additional data file.

Supplemental Information 8 BMP-2 ELISA results

Click here for additional data file.

Supplemental Information 9 Col-2 ELISA results

Click here for additional data file.

Supplemental Information 10 Integrin alfa 3 qPCR results

Click here for additional data file.

Supplemental Information 11 Integrin alfa 4 qPCR results

Click here for additional data file.

Supplemental Information 12 Integrin alfa 5 qPCR results

Click here for additional data file.

Supplemental Information 13 Integrin alfa V qPCR results

Click here for additional data file.

Supplemental Information 14 Integrin Beta 1 qPCR results

Click here for additional data file.

Supplemental Information 15 Integrin beta 3 qPCR results

Click here for additional data file.

Supplemental Information 16 Integrin beta 5 qPCR results

Click here for additional data file.

Supplemental Information 17 PPAR-gamma qPCR results

Click here for additional data file.

Supplemental Information 18 Adiponectin qPCR results

Click here for additional data file.

Supplemental Information 19 LEPTIN qPCR results

Click here for additional data file.

Supplemental Information 20 Average diameters of nodules

Click here for additional data file.

Supplemental Information 21 proliferation factor

Click here for additional data file.

Supplemental Information 22 Population doubling time

Click here for additional data file.

We thank Marta Jeleń for assistance during in vitro work and cell vibration generator prototype preparation.

Additional Information and Declarations

Competing Interests

Author Contributions

Human Ethics

Data Availability

The authors declare there are no competing interests regarding the publication of this paper.

Krzysztof Marycz conceived and designed the experiments, analyzed the data, contributed reagents/materials/analysis tools, wrote the paper, reviewed drafts of the paper.

Daniel Lewandowski conceived and designed the experiments, performed the experiments, wrote the paper.

Krzysztof A. Tomaszewski analyzed the data, contributed reagents/materials/analysis tools, wrote the paper, reviewed drafts of the paper.

Brandon Michael Henry analyzed the data, wrote the paper, reviewed drafts of the paper.

Edward B. Golec contributed reagents/materials/analysis tools, reviewed drafts of the paper.

Monika Marędziak performed the experiments, analyzed the data, wrote the paper, prepared figures and/or tables.

The following information was supplied relating to ethical approvals (i.e., approving body and any reference numbers):

The study was approved by the local bioethics committee of Wroclaw Medical University, Poland (number KB-177/2014). Written informed consent was obtained from each patient prior to tissue collection during total hip arthroplasty. This study adhered to the Helsinki Declaration (1964) and its later amendments.

The following information was supplied regarding data availability:

The raw data is provided as Supplemental Dataset files. The results were generated via GraphPad Prism and exported to csv files.

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
