# Peer review of "Low-frequency, low-magnitude vibrations (LFLM) enhances chondrogenic differentiation potential of human adipose derived mesenchymal stromal stem cells (hASCs)"

_PeerJ, doi:10.7717/peerj.1637_

## Round 0.1 · original submission · Major Revisions

Dear Dr. Marycz,

The reviewers have provided constructive comments and suggestions for how to strengthen the manuscript. Please respond to the comments, particularly focusing on the need for additional details such as the characterization of the cells and requested quantification, showing the requested data, improving the contextual discussion of your work by including the suggested references and limiting your interpretations to the definitive results.

Reviewer 1 ·

Basic reporting

Authors test the effects of low frequency low magnitude (LFLM) stimulation on the in-vitro differentiation potential of adipose derived human MSCs (ASC). Author’s test the efficacy of LFLM signal at 25, 35 and 45 Hz frequencies using a 0.3g acceleration magnitude and conclude that LFLM affects ASC proliferation, chondrogenesis and adipogenesis as well as associated gene expressions, protein secretion and morphology.

However, throughout the manuscript authors consistently use blanket statements about LFLM effects that sometimes do not reflect the presented data adequately. Further, in many cases authors makes statements without appropriate referencing references. Reviewer believes that the current manuscript can be accepted for publication only after major revisions as outlined below.

1. Line 52, please provide a reference for NSAIDs causing chondronecrosis.
2. Line 61, please provide references for ASCs secreting microvesicles containing important regenerative molecules.
3. Lines 62-64, Revise the sentence, it is not clear if MSCs or Osteocytes secrete microvesicles.
4. Line 78, please provide references for LFLM applied <1g and between 20-100Hz.
5. Lines 78-82, to a large degree authors do not cite any in vitro studies that deal with LFLM, in fact only “cell-level” study they site uses >3g acceleration magnitude. There are many recent studies that showed that both MSCs and ASCs can respond to low magnitude vibrations and regulate their adipogenesis and osteogenesis and cellular structure accordingly (Uzer et al Stem Cells 2015, Sen et al J.Biomech 2011, Prè et al Bone 2011, Uzer et al J biomech 2013 ) for more detailed literature review authors are encouraged to explore review articles (Chan et al Curr Osteoporos Rep. 2013, Rubin et al Gene 2006) and should include appropriate references throughout the manuscript.
6. Line 85, please provide a references for integrins being upregulated by various biomechanical signals.
7. Line 93, authors talk about bone recovery after fracture but the reference do not include any fracture. Please include appropriate references ( including but not limited to Goodship et al J Orthop Res. 2009)

Experimental design

Experimental design is appropriate and methods are sound. However data about proliferation and the added value of Fig.4B is not clear. Some comments regarding the results section are given below.

8. Regarding the “cell vibration generator” did authors measured motions of the plate at different locations as partial support may cause secondary – higher frequency-- vibrations on the cell culture plate. What was the accuracy of the displacement measurements, this would be important to understand the ability to measure secondary vibrations. Authors should provide the peak displacement magnitudes for each frequency preferably in a table format. It is also not clear 0.3g is peak or peak to peak acceleration magnitude.
9. Authors use/discuss the data from 2D and 3D cultures interchangeably throughout the manuscript. Especially considering the radical differences between cell morphology in 2D and 3D environments it would be expected that extracellular proteins and integrins would have different behavior. This is probably reflected in the Col-2 data where 2D cultures show an increase (35Hz) with LFLM but no corresponding increase in Col-2 protein in 3D cultures. Authors should make this distinction early on and discuss this limitations in the discussion section.
10. Authors indicate that 25Hz LFLM group reached the highest PF after 2 days however the earliest data point provided is day 2 as well. How many cells has been plated at day 0? Please include a data point for day 0.
11. Regarding the PF data, none of the graphs show significance (asterix) but the text talks about significance, either add this information into the figure or not say it in the text.
12. In Figure 4 DAPI staining (which stains DNA) is visible in whole cell, which makes it very hard to judge about cell structure.
13. Additionally, authors also state that ”The highest difference between collagen type II and type I was observed in the cultures stimulated with 25 and 35 Hz vibrations, and in the control culture” However, neither 25Hz nor 35Hz groups show increased Col-1 or Col-2 levels as only significant (and biggest) difference is observed at 45Hz.
14. Regarding the cell morphology of ASCs cultured under adipogenic conditions, what was the sample size used for each group and how cell size was determined.

Validity of the findings

Findings appear to be valid, however authors over-interpret some of the data, some comments regarding the discussion section is below.

15. Line 412, not clear what ECM of high biomechanical properties mean.
16. Line 415, please refer to the figure that shows cytoskeletal formation.
17. Sentence at line 420 and 426 indicates that there was an increase in Col-2 production in response to vibration. Fig.5 clearly indicates a no change in Col-1 and Col-2 in response to 25 and 35Hz vibrations. Authors encouraged to refer to appropriate figures or delete this part of the discussion.
18. Please provide a reference for the sentence “Moreover, SOX9 determines functions of RUNX2 and exerts a dominant function over RUNX2 in mesenchymal precursors.”
19. Discussion between lines 441 and 450 do not reflect the data please revise.

·

Basic reporting

The authors have studied if low-frequency, low-magnitude vibration could enhance chondrogenic differentiation potential of hASCs and also inhibit adipogenic differentiation. The authors utilize an in-house-built device for applying vibration stimulation. This is an interesting topic that has not yet been extensively studied specifically in the context of chondrogenesis, and hence this study does contribute to filling a gap in the field. Overall the article is well-written and logical. It has a sufficient introduction, discusses the current results in comparison to previous findings and refers to relevant literature.

Experimental design

The research question is clearly defined and relevant. The authors have conducted various experiments to find out the effect of LMLF on hASCs, particularly chondrogenic and adipogenic differentiation.

Validity of the findings

Major issues:

1) It is unclear how many donors (i.e. cells from different donors) were actually used for conducting the experiments. It is mentioned on row 270 that n =3 or more. Does it mean 3 repetitions with one donor or conducting the experiments with cells from 3 donors?
2) Characterization of hASCs is not clear. FACS data should be presented as mean % and SEM or SD (mean from all the donors used for the experiments). From this data it should be evaluated if the cells meet the ISCT guidelines (<95% for CD90, CD73 and CD105 etc.).
3) It is not described in the Materials and Methods part how exactly the authors quantified:
- Percentage of Safranin staining absorption (fig 6)
- Average diameter of nodules (fig.6) (e.g. average from how many nodules).
- Percentage of Oil Red O staining absorption (fig. 10)
- Average diameter of adipocytes (fig. 10)
4) This is a major problem (point 3), because quantification of staining/nodules is often semi-quantitative and therefore prone to subjectivity of the researcher. If this would be the case here, the conclusion that vibration at 35 Hz enhances chondrogenic differentiation of hASCs would be mostly based on qRT-PCR results. Furthermore, if the qRT-PCR results are based on repetitions with cells from only one donor, I would not consider this a conclusive finding.
5) Why did you use 3D pellet culture for ELISA analysis and 2D for other analyses? Can you explain in more detail in Materials and Methods how the vibration stimulation was conducted for the 3D pellets? You should also mention how long you allowed the cells to attach in 2D before starting the vibration stimulation and how long you allowed the pellet to form in 3D before vibration stimulation.

Minor issues:

6) Row 134: Vibration was conducted for 14 consecutive days. But you also used 21 days of culture for chondrogenic differentiation with vibration in 3D? If so, it should be mentioned here as well.
7) Row 238-239: Check the title, it is not clear.
8) Row 206: Correct Almar to Alamar.
9) Figure 8: Correct the title (adipogenesis).

---

## Round 0.2 · accepted · Accept

Dear Dr. Marycz,

Your revision has addressed the concerns of the reviewers and we believe that the manuscript will provide important novel insights into the regulation of adipose MSC differentiation.

Reviewer 1 ·

Basic reporting

No Comments

Experimental design

No Comments

Validity of the findings

No Comments

Additional comments

The revised version of the manuscript was well-documented with introducing reviewers comments. My major concern in the first version was largely solved. I wish to congratulate the authors for this well-written and interesting paper.

·

Basic reporting

The authors have addressed the comments adequately.

Experimental design

No comments.

Validity of the findings

No comments.